# Jellyfish blooms—an overlooked hotspot and potential vector for the transmission of antimicrobial resistance in marine environments

Alan X. Elena,[1] Neža Orel,[2] Peiju Fang,[1,3] Gerhard J. Herndl,[4,5,6] Thomas U. Berendonk,[1] Tinkara Tinta,[2,4] Uli Klümper[1]

**ABSTRACT**    Gelatinous zooplankton (GZ) represents an important component of marine food webs, capable of generating massive blooms with severe environmental impact. When these blooms collapse, considerable amounts of organic matter (GZ-OM) either sink to the seafloor or can be introduced into the ocean's interior, promoting bacterial growth and providing a colonizable surface for microbial interactions. We hypothesized that GZ-OM is an overlooked marine hotspot for transmitting antimicrobial resistance genes (ARGs). To test this, we first re-analyzed metagenomes from two previous studies that experimentally evolved marine microbial communities in the presence and absence of OM from *Aurelia aurita* and *Mnemiopsis leidyi* recovered from bloom events and thereafter performed additional time-resolved GZ-OM degradation experiments to improve sample size and statistical power of our analysis. We analyzed these communities for composition, ARG, and mobile genetic element (MGE) content. Communities exposed to GZ-OM displayed up to fourfold increased relative ARG and up to 10-fold increased MGE abundance per 16S rRNA gene copy compared to the controls. This pattern was consistent across ARG and MGE classes and independent of the GZ species, indicating that nutrient influx and colonizable surfaces drive these changes. Potential ARG carriers included genera containing potential pathogens raising concerns of ARG transfer to pathogenic strains. *Vibrio* was pinpointed as a key player associated with elevated ARGs and MGEs. Whole-genome sequencing of a *Vibrio* isolate revealed the genetic capability for ARG mobilization and transfer. This study establishes the first link between two emerging issues of marine coastal zones, jellyfish blooms and ARG spread, both likely increasing with future ocean change. Hence, jellyfish blooms are a quintessential "One Health" issue where decreasing environmental health directly impacts human health.

**IMPORTANCE**    Jellyfish blooms are, in the context of human health, often seen as mainly problematic for oceanic bathing. Here we demonstrate that they may also play a critical role as marine environmental hotspots for the transmission of antimicrobial resistance (AMR). This study employed (re-)analyses of microcosm experiments to investigate how particulate organic matter introduced to the ocean from collapsed jellyfish blooms, specifically *Aurelia aurita* and *Mnemiopsis leidyi*, can significantly increase the presence of antimicrobial resistance genes and mobile genetic elements in marine microbial communities by up to one order of magnitude. By providing abundant nutrients and surfaces for bacterial colonization, organic matter from these blooms enhances ARG proliferation, including transfer to and mobility in potentially pathogenic bacteria like *Vibrio*. Understanding this connection highlights the importance of monitoring jellyfish blooms as part of marine health assessments and developing strategies to mitigate the spread of AMR in coastal ecosystems.

**Peer Reviewer** Tamar Guy-Haim, Israel Oceanographic and Limnological Research Institute, Haifa, Israel

Address correspondence to Uli Klümper, uli.kluemper@tu-dresden.de, or Tinkara Tinta, tinkara.tinta@nib.si.

The authors declare no conflict of interest.

See the funding table on p. 16.

**KEYWORDS** antimicrobial resistance, jellyfish blooms, gelatinous zooplankton, marine microbiomes, horizontal gene transfer, *Mnemiopsis leidyi*, *Aurelia aurita*, *Vibrio*

Gelatinous zooplankton represents an important component of marine food webs inhabiting tropical to polar marine ecosystems. They represent ~30% of the total biovolume, corresponding to 8%–9% of the globally stored carbon in planktonic communities (1). The most common groups among marine gelatinous zooplankton include medusae (jellyfish), ctenophores (comb jellies), salps, and chaetognaths. Within the context of climate change, given future ocean projections and considering the adaptability of gelatinous zooplankton to a wide range of environmental conditions these organisms will likely increasingly dominate planktonic marine ecosystems, leading to significant changes in the ocean's carbon cycle (1–3). An increase in gelatinous zooplankton abundance has already been recorded worldwide, particularly in anthropogenically impacted coastal areas, which is threatening marine ecosystem health and services (2, 4).

Due to a combination of life-history traits and low metabolic requirements certain gelatinous zooplankton species (e.g., *Mnemiopsis leidyi, Aurelia aurita*) (5, 6) are capable of generating massive gelatinous zooplankton blooms (hereinafter GZ-blooms), representing an important perturbation to marine ecosystems (7–9). These GZ-blooms are often followed by a sudden collapse of the entire population resulting in a large influx of gelatinous zooplankton detrital organic matter (hereinafter GZ-OM) that transforms the ambient seawater organic matter pool by releasing considerable amounts of bloom-specific particulate and dissolved organic and inorganic matter compounds (10–12). GZ-OM can then be degraded and consumed at different rates in a cascade by specific microbial assemblages, dominated by copiotrophic bacterial lineages, with consistent metabolic fingerprints (7, 9, 13). In this study, we hypothesize that the GZ-OM introduced into ocean ecosystems upon the decay of GZ-bloom events serves as a yet overlooked hotspot for transmitting antimicrobial resistance genes (ARGs) in marine environments.

Antimicrobial resistance (AMR) and the spread of ARGs is one of the major global health challenges (14) with globally already 4.71 million deaths associated and 1.14 million deaths directly attributable to bacterial AMR in 2021 (15). To mitigate the predicted future rise in these numbers, it is important to understand AMR evolution, selection, and transmission within and across all interconnected "One Health" compartments (humans, animals, and the environment) (16, 17). Especially, understanding the biotic and abiotic drivers underlying this spread is crucial to creating targeted intervention measures (18, 19). With abundance of ARGs increasing in many ecosystems due to anthropogenic activities (20, 21), marine ecosystems and their microbiomes are no exception (16, 22). In particular, coastal zones as a likely entry point of ARG-carrying microbes from, for example, wastewater effluents to the marine environments are in the spotlight as they provide exposure points to humans who are using them recreationally (23). For example, increased colonization of marine surfers with AMR bacteria has previously been proven (24). Moreover, various pollutants of marine ecosystems ranging from chemicals to microplastics can contribute to the spread of AMR in non-coastal marine ecosystems which can accumulate in marine animal microbiomes and subsequently through the food chain be conveyed back to terrestrial animals and humans (25). Yet, potential links between these two emerging issues of anthropogenically impacted marine zones, bloom-forming gelatinous zooplankton species, and ARG have not been explored.

One of the most important ecological mechanisms underlying this spread of ARGs is the conjugative transfer of ARG-encoding mobile genetic elements (MGEs) such as plasmids (26, 27). These conjugative plasmids can spread between closely related bacteria but also be transferred to phylogenetically distant bacterial groups (28–31). Plasmid transfer rates in aquatic environments are particularly elevated when bacterial abundances and activity are high, ensuring high bacterial encounter rates such as in

biofilm formed on the surfaces of particles (32–34). Both these factors are given during the decay of GZ-bloom events as the released GZ-OM represents an abundant substrate to promote bacterial growth and copious colonizable surfaces for interactions. Furthermore, microbial degraders of GZ-OM, potentially enriched in AMR, could hitchhike on these organic particle surfaces by drifting with ocean currents over long distances into the oceanic interior (35, 36) and/or connect the microbiomes of upper and bottom ocean layers upon sinking to the ocean floor (37, 38). Thus, GZ-bloom events could provide a yet overlooked hot spot for the spread of AMR and a potential vector of AMR/ARG transmission in marine environments.

Consequently, to address the hypothesis that GZ-OM degrading microbial communities provide a hot spot for the spread of AMR, we re-analyzed existing metagenomic data sets (7, 13) from microcosm experiments previously conducted and performed and analyzed new, time-resolved, and replicated GZ-OM degradation experiments to increase the statistical power of the analysis. This provided insights into how the microbial degradation of biomass from different bloom-forming gelatinous zooplankton species affects the abundance, diversity, and dynamics of AMR and MGEs.

## MATERIALS AND METHODS

### Re-analyzed metagenomic data sets

The initial metagenomic data sets used in this study originate from previously conducted experiments on the degradation of gelatinous zooplankton-derived organic matter (hereinafter GZ-OM) described in detail in Tinta et al. (2020, 2023) and Fadeev et al. (2024) (7, 12, 13). Briefly, we conducted short-term microcosm evolution experiments to simulate the scenario potentially experienced by the coastal pelagic microbiome after the decay of GZ-blooms. The first set of experiments was conducted using biomass of the cosmopolitan scyphozoan jellyfish, *Aurelia aurita* s.l. (13). The second set of experiments was conducted using biomass of the lobed ctenophore *Mnemiopsis leidyi*, one of the most notorious marine-invasive species (7). Both species form massive aggregations in numerous ecosystems around the world, causing a threat to marine ecosystem services. For each evolution experiment, we filled six 10 L borosilicate glass microcosms with 0.2 µm filtered aged seawater (ASW), which was inoculated with a 1.2 µm prefiltered coastal microbial community collected in near-surface waters of the northern Adriatic Sea in a ratio of ASW:bacterial inoculum of 9:1. Based on the average abundance of gelatinous zooplankton per m$^3$ during typical GZ-bloom conditions in our study coastal ecosystem—the northern Adriatic Sea—three experimental microcosms representing the GZ-OM treatment, received 100 mg L$^{-1}$ of GZ-OM (either *A. aurita* or *M. leidyi*). Three microcosms with no GZ-OM amendment served as the control treatment. All microcosms were incubated in the dark at *in situ* temperature (~24°C) and mixed gently before sampling. Bacterial community dynamics (abundance, production, and single-cell metabolic activities) and organic/inorganic nutrients are described in detail in Tinta et al. (*A. aurita* data set)(13) and Fadeev et al. (*M. leidyi* data set)(7). At the same time, the microbial inoculum and microbial communities at the peak of bacterial abundance were sampled after 21 h for *M. leidyi* and 32 h for *A. aurita* for microbial metagenome and metaproteome (endo- and exo-fraction) analysis from each of the replicate microcosms per treatment. DNA was extracted according to Angel et al. (39) with minor modifications for GZ-OM as described in Tinta et al. (12). Extracted DNA from each replicate was pooled in equimolar amounts before metagenomic sequencing, resulting in one metagenome data set per treatment. This approach provided early insights into metabolic networks operated by microbial degraders of GZ-OM (7, 13) and was here re-analyzed for AMR and MGE content.

### Time-resolved and replicated experiment of *M. leidyi* GZ-OM degradation

To complement the already available data sets, we performed a replicated experimental design of the *M. leidyi*-OM microcosms as described in Fadeev et al. (7) and above. To

increase the statistical validity of the insights gained, we produced metagenomes of the microbial inoculum, and each of the triplicate microbial communities in either the presence or absence of *M. leidyi* GZ-OM sampled after 43 and 67 hours.

## DNA extraction and metagenomic sequencing

Metagenomic DNA library preparation and sequencing were performed as described previously (12, 13). Raw reads were deposited at NCBI under the accession number PRJNA633735. The obtained raw reads were assessed quality-wise using FastQC (40). Downstream quality processing was performed using BBTools (41), and the quality trimming was performed using bbduk, employing a Phred-based algorithm and a cut-off of 25 and a right and left trimming approach. For the detection of potential phiX phage residual contamination, bbduk was used using a k-mer size of 31 and the integrated phiX fasta sequence.

## Metagenomic analysis

The taxonomic composition of each sample was assessed using Kraken2 (42) with a k-mer-based approach. For this, we used the Kraken2 PlusPF database and adjusted the confidence to 0.175, a value optimized using a set of high-quality mock communities (SRR8073716 (43) and SRP436666 (44)). To account for the known underestimation of taxonomic assignments, abundance re-classification through a naïve Bayesian approach using Bracken was carried out (45). Antimicrobial resistance genes and the 16S rRNA gene content in the metagenomes were predicted in each clean metagenomic data set using the ARGs-OAP V2.0 (46) in which a specific ARG database (Structured Antibiotic Resistance Genes [SARG]) constructed from RefSeq ARG hidden Markov Models is used as a reference. Metagenomic reads are aligned to the reference using BLASTx and a 90% identity cutoff. The abundance of reads matching a certain ARG is recalculated taking into consideration the RefSeq sequence length, therefore accounting for uneven coverage and potential over-estimation. The 16S rRNA gene content is similarly estimated and then used to normalize the ARG abundance (46, 47). For mobile genetic element (IS and replicons) relative abundance estimation, a similar approach was used: In these cases, the reference databases were downloaded from ISFinder (48) and Plasmid-Finder (49) and formatted accordingly using ARGs-OAP's (46) make-db command.

## *Vibrio splendidus* isolation and whole-genome analysis

*Vibrio* was identified as a key bacterial group involved in GZ-OM degradation and the spread of AMR in this study. No isolation of bacteria was performed during the experiments described in (7, 13) or in this manuscript. However, we had access to a *Vibrio* A06 isolate from a previous GZ-OM degradation experiment with *Aurelia aurita* biomass described in detail in reference (50). Upon whole-genome analysis *Vibrio* A06 was identified to be closely related to organisms within the same taxonomic groups present in our metagenomic data sets. Consequently, the *Vibrio* A06 isolate's whole-genome sequence was analyzed in this study to gain further genomic insights into the role *Vibrio* might play in the spread of AMR during GZ-OM degradation:

For bacterial isolation of this strain, 100 µL from GZ-OM microcosms (50) was spread on modified ZoBell solid agar media and incubated in the dark at 21°C by gently agitating for 48 h. Single colonies were clean streaked once, inoculated into ZoBell liquid medium, and incubated in the dark at 21°C for 24 h. Furthermore, the bacterial isolate was stored at the culture collection of the Marine Biology Station Piran, Slovenia (in 30% glycerol at −80°C). The isolate was regrown for this study from cryo-preserved stock on ZoBell agar plates in the dark at 24°C for 72 h. Then a single colony was inoculated into 6 mL of ZoBell liquid medium, and incubated at room temperature in the dark on a shaker. Four 1 mL replicates of the liquid culture were pelleted by centrifugation at $4,000 \times g$ for 3 min and bacterial pellets were shipped on dry ice to the sequencing facility (Microsynth AG, Balgach, Switzerland) where high molecular weight

DNA was extracted. The DNA was then sequenced using the long-read MinION ONT (Oxford Nanopore Technologies, Oxford, United Kingdom) technique, complemented by short-read paired-end (2 × 75 bp) sequencing on Illumina NextSeq (Illumina, San Diego, CA, USA). Raw long and short reads were quality trimmed using Filtlong V0.2.1 and Trimmomatic V0.39 (51) and used to generate a high-quality hybrid assembly with Unicycler V0.4.8 (52). A total of three contigs were assembled and annotated using the RAST server (53). The taxonomic assignment of isolate *Vibrio* A06 was first carried out using the classical 16S rRNA DNA approach. For this, 16S rRNA gene sequences were extracted using Barrnap V0.9 (54), concatenated and aligned to 69 available complete *Vibrio* genomes (Taxid: 662) retrieved from the NCBI database and Enterobase (55) using clustalW (56). Previous studies indicate that *Vibrio* species can be more accurately determined by comparison of the heat shock protein 60 (HSP60)(57, 58). Therefore, as an additional identification strategy, we compared the HSP60 sequence of A06 using a similar approach as for the 16S rRNA gene. ARGs, replicon-associated structures, and virulence factors were screened using the ResFinder, PlasmidFinder, and VirulenceFactor databases (48, 59, 60).

## Statistical assessment

Statistical evaluations were carried out in R V4.2.2 (61). Differences in ARG, MGE, or bacterial group abundances between sample types were assessed using the non-parametric Friedman's Test. Dynamics of ARG and MGE abundances over time were assessed using Pearson correlation tests. For analysis of microbial community and ARG diversity, predicted OTU abundances were transformed using the Hellinger method, and Euclidean distances between samples were calculated using the Vegan package in R (62). Euclidean distances based on relative ARG abundance between each sample were calculated without previous transformation. The correlation between OTU and ARG ordinations was computed using a symmetric Procrustes approach with ordination plots created using the ggplot2 package (63). Statistical differences between different groups of samples in the plots were calculated using the analysis of molecular variance (AMOVA) (64). Throughout, statistical significance was assigned at $P < 0.05$. A pairwise correlation analysis between the ARG and bacterial genera abundances based on Spearman rank correlation was assessed to explore potential ARG-host relationships. Only positive correlations showing coefficients $\rho > 0.75$ were considered significant at $P < 0.05$ after Benjamini-Hochberg correction for multiple testing. The correlations were displayed through network analysis, performed using the picante package (65) based on the significant Spearman rank correlations. The network was visualized using the open-source software Gephi v8.2 (66).

## RESULTS

### Antimicrobial resistance dynamics during microbial degradation of gelatinous zooplankton organic matter

To explore whether GZ-OM following the collapse of GZ-blooms could result in the proliferation of AMR, we first reanalyzed metagenomes from lab-based microcosm degradation experiments of *Mnemiopsis leidyi* (13) and *Aurelia aurita* (7). Microcosms were seeded with a coastal marine microbial community in the presence and absence of *M. leidyi* or *A. aurita* OM and were terminated after the microbial community reached the late exponential growth phase.

In the *M. leidyi* experiment, the total relative abundance of ARGs decreased in the absence of GZ-OM from $1.61 \times 10^{-2}$ ARGs per copy of the 16S rRNA gene in the coastal marine microbiome to $8.18 \times 10^{-3}$ ARGs/16S (Fig. 1A). By contrast, the microbiome exposed to GZ-OM was highly enriched in ARGs with a final relative abundance of $5.46 \times 10^{-2}$ ARGs/16S within less than 2 days of exposure (Fig. 1A). The main prevalent antibiotic classes ($>10^{-3}$ ARGs/16S) these ARGs conferred resistance to were polymyxins, tetracyclines, quinolones, trimethoprim, fosfomycin, and chloramphenicol (Fig. 1B). The

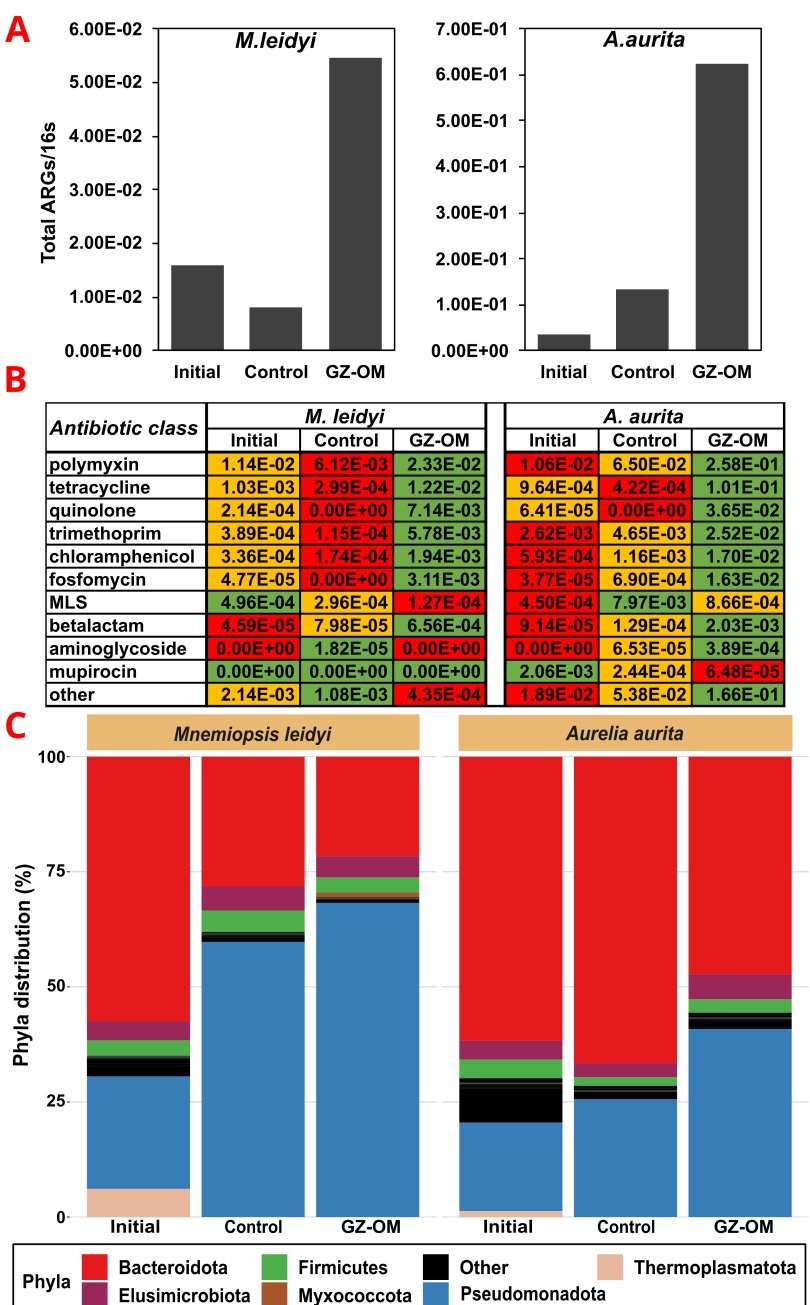

**FIG 1** Characterization of the resistome and the microbiome of the initial microbial community as well as those from microcosm experiments in the presence and absence of *Mnemiopsis leidyi* or *Aurelia aurita* GZ-OM after 21 hours for *M. leidyi* and 32 hours for *A. aurita*. (A) Relative abundance of total ARGs. (B) Relative abundance of ARGs based on antibiotic classes to which they confer resistance, color-coded for highest (green), intermediate (yellow), and lowest (red) abundance in the initial, control, and GZ-OM treatment community. (C) Microbial community composition in the microcosm on the phylum level, phyla with abundance below 1% are grouped as others.

observed trend for total ARGs was also consistent at the antibiotic class level: For the majority of detected antibiotic classes (7 out of 10), the relative abundance of the detected ARGs conferring resistance to that class was decreased in the control treatment compared to the initial community (Fig. 1B). On the contrary, for the GZ-OM treatment, a significant increase in relative ARG abundance across antibiotic classes was observed

when compared to either the original community (7 out of 10 ARG classes increase) or the control treatment (7 out of 10 ARG classes) (Fig. 1B).

Similar trends were observed in the *A. aurelia* experiment, where a strong increase in the total relative abundance of ARGs from $3.64 \times 10^{-2}$ in the initial microbiome to $6.23 \times 10^{-1}$ ARGs/16S in the presence of jellyfish biomass was observed. This accounted for a >4-fold increase compared to the no jellyfish control $1.34 \times 10^{-1}$ ARGs/16S (Fig. 1A). The observed trend remained again consistent at the antibiotic class level with similar antibiotic classes (polymyxins, tetracyclines, quinolones, trimethoprim, chloramphenicol, and fosfomycin) dominating the resistome: For the majority of ARG classes, relative abundances were consistently increased in the GZ-OM treatment compared to the initial microbiome (10 out of 11 ARG classes) and the control treatment (9 out of 11 ARG classes) (Fig. 1B). In summary, we observed early trends of AMR increasing during GZ-OM degradation independent of the GZ-OM origin species and across antibiotic classes.

## Microbial players responsible for GZ-OM degradation and involved in the proliferation of AMR

We next identified the main bacterial groups responsible for GZ-OM degradation and hence likely involved in the observed increase in AMR in the degrading communities. In both experiments (*M. leidyi* and *A. aurita*), the initial microbial communities were mainly dominated by *Bacteroidota* and *Pseudomonadota*. However, a clear shift toward a higher abundance of *Pseudomonadota* at the expense of *Bacteroidota* was observed in both the control as well as the GZ-OM treatments (Fig. 1C). This shift was far more pronounced in those microcosms exposed to either of the two types of GZ-OM (Fig. 1C). We identified candidate genera that might have contributed to the observed increase in AMR as they displayed a high abundance (>0.1% rel. abundance) and were found at >5-fold increased abundance in the GZ-OM compared to the control treatment. For both types of GZ-OM, these included well-known GZ-OM degraders (*Pseudoalteromonas, Vibrio, Alteromonas*) (7, 13), but also genera that contain human or animal pathogens (*Enterobacter, Escherichia-Shigella, Vibrio, Pajaroellobacter, Francisella*) (67). In addition, *Oleiphilus, Anaerosinus, Acinetobacter,* and *Arsenophonus* increased in abundance in the *M. leidyi* experiment. In summary, similar microbial genera dominated both *M. leidyi* and *A. aurita* GZ-OM degradation, indicating that despite the missing replication in this initial data set, the observed trends are generalizable for biomass degradation of diverse GZ species. Still, as in these original studies we only operate with a single metagenome data set per treatment, and at a single time point, insights gained remained limited. Thus, an additional GZ-OM degradation experiment with replicates and samples taken over different time points was performed and analyzed.

## Antimicrobial resistance genes are consistently enriched in microbial communities degrading *Mnemiopsis leidyi* GZ-OM

To gain statistically valid insights into whether degradation of GZ-OM indeed increases AMR proliferation in the marine microbiome, we analyzed metagenomes from the newly performed *M. leidyi*-OM degradation experiment with the appropriate replication, sampling, and sequencing at three time points representing the microbial inoculum ($T_{0h}$), the microbial community at the late exponential phase of their growth ($T_{43h}$), and the microbial community after reaching stationary growth phase ($T_{67h}$). Again, total ARG relative abundance increased in the GZ-OM treatment from $0.91 \times 10^{-2}$ ($T_{0h}$) to $3.85 \pm 1.85 \times 10^{-2}$ ($T_{67h}$) ARGs per 16S rRNA gene with a significant increase rate of $4.9 \times 10^{-4}$ ARGs/16S per hour (R = 0.68, $P = 0.048$, Pearson correlation) (Fig. 2A).

However, no significant difference in ARG levels over time (m = $2.8 \times 10^{-4}$ ARGs/16S per hour, R = 0.65, $P = 0.121$) (Fig. 2A) was detected in the control treatment without GZ-OM addition. This increase at the ARG class level displayed a clear effect of GZ-OM especially for tetracycline and fluoroquinolone ARGs. Tetracycline ARGs significantly increased in proportion from initially 4.1% of the total ARGs to $20.0 \pm 13.2\%$ in GZ-OM treatments [$P = 0.0498$, Friedman's $\chi^2(2) = 6$, Friedman's test]. By contrast, the proportion

of tetracycline ARGs remained rather stable in the control treatment [2.8 ± 2.8%, $P$ = 0.71, Friedman's $\chi^2(2)$ =0.67] (Fig. 2B). Similarly, fluoroquinolone resistance genes increased from 1.1% to 12.7 ± 9.5% in GZ-OM treatments [$P$ = 0.0498, Friedman's $\chi^2(2)$ =6] while remaining stable in the control at 1.5 ± 2.1% [$P$ = 1.0, Friedman's $\chi^2(2)$ =0] (Fig. 2B). This increase in proportion was mirrored in the relative abundances increasing by more than one order of magnitude from initially $3.79 \times 10^{-4}$ to $9.87 \pm 8.45 \times 10^{-3}$ tetracycline ARGs/16S and from $9.98 \times 10^{-5}$ to $6.48 \pm 5.29 \times 10^{-3}$ fluoroquinolone ARGs/16S which was significant for fluoroquinolones [$P$ = 0.0498, Friedman's $\chi^2(2)$ =6], while not statistically significant but strongly indicated for tetracycline [$P$ = 0.097, Friedman's $\chi^2(2)$ =4.67]. For the remaining antibiotic classes, the shifts were smaller and remained below one order of magnitude in relative abundance.

## Mobile genetic elements are consistently enriched in GZ-OM degrading microbial communities

As increases in AMR are of particular risk if associated with a rise in MGEs that can facilitate their transfer to pathogenic strains, we similarly analyzed the MGE content of the metagenomic samples with a particular focus on insertion sequences (IS). While the number of detected IS families remained at a consistent level between 18 and 23 independent of any treatment, the relative IS abundance per 16S rRNA gene mirrored that of ARGs. In both initial data sets IS relative abundance was increased in the GZ-OM treatment (*M. leidyi*: $3.41 \times 10^{-1}$ IS/16S, *A. aurita*: $2.83 \times 10^{-1}$ IS/16S, $n$ = 1) compared to their respective control (*M. leidyi*: $1.97 \times 10^{-2}$ IS/16S, *A. aurita*: $1.98 \times 10^{-2}$ IS/16S, $n$ = 1) (Fig. 3A). Similarly, in the time-resolved experiment, a statistically significant increase in IS relative abundance was observed for the GZ-OM treatment at both timepoints [43 h: $1.21 \pm 0.96 \times 10^{-1}$ vs. $3.67 \pm 2.89 \times 10^{-2}$ IS/16S, 67 h: $1.72 \pm 1.40 \times 10^{-1}$ vs. $2.82 \pm 1.91 \times 10^{-2}$ IS/16S, $P$ = 0.0498, Friedman's $\chi^2(2)$ =6] (Fig. 3A). The observed effect was largely consistent across the different individual IS families, however, a particularly significant increase in the proportion of the IS*110* family, an IS family reported to regularly form ARG-carrying transposons (68), in the *M. leidyi*-OM (28.6 ± 6.7%) compared to the control treatment (8.1 ± 2.4%, $P$ < 0.001, $t$-test) stood out (Fig. 3B). Within this IS*110* family, we detected 36 individual IS out of which IS*Visp2*, IS*Visp6*, IS*Spi5*, IS*Ptu2*, IS*Spi2*, and IS*Cps8* were significantly enriched in the *M. leidyi*-OM treatment. Similarly, in the *A. aurita*-OM treatment, we observed 27 individual IS belonging to the IS*110* family. Again, IS*Visp6*, IS*Visp2*, IS*Ptu2*, IS*Spi5*, and IS*Cps8* were almost exclusively associated with GZ-OM. In general, IS*Visp2* and IS*Visp6*, native to the *Vibrio* species *Vibrio splendidus*, were the most abundant of the IS*110* members detected in both, the *M. leiydi*-OM and the *A. aurita*-OM treatment in our experiments. This places *Vibrio* not only as a relevant GZ-OM degrader but also as a potential genus involved in horizontal ARG dissemination.

## Increase in antimicrobial resistance during *Mnemiopsis leidyi* biomass degradation is related to the microbial community composition

As GZ biomass is unlikely to contain large amounts of selective agents that are responsible for the observed increase in ARGs and MGEs, a more likely explanation is a shift in microbial community composition during the degradation process. Indeed, for both the GZ-OM and the control treatment, a clear shift in community composition at the genus level was observed when comparing the microcosm communities to the initial community composition ($P$ < 0.05, AMOVA) (Fig. 4A). However, GZ-OM treatment communities consistently grouped further apart from the initial community (0.627 ± 0.029 average Hellinger transformed taxonomic distance) than the control treatment (0.534 ± 0.069, $P$ = 0.011, ANOVA) (Fig. 4A).

Moreover, the control and GZ treatments were grouped apart from each other ($P$ < 0.05, AMOVA). Still, no clear effect of the sampling time between $T_{43h}$ and $T_{67h}$ could be observed for either the GZ-OM or control samples (all $P$ > 0.05), indicating that the shifts in community composition appeared early during GZ-OM degradation. The individual ARG diversity between GZ-OM-treatment, control treatment, and initial

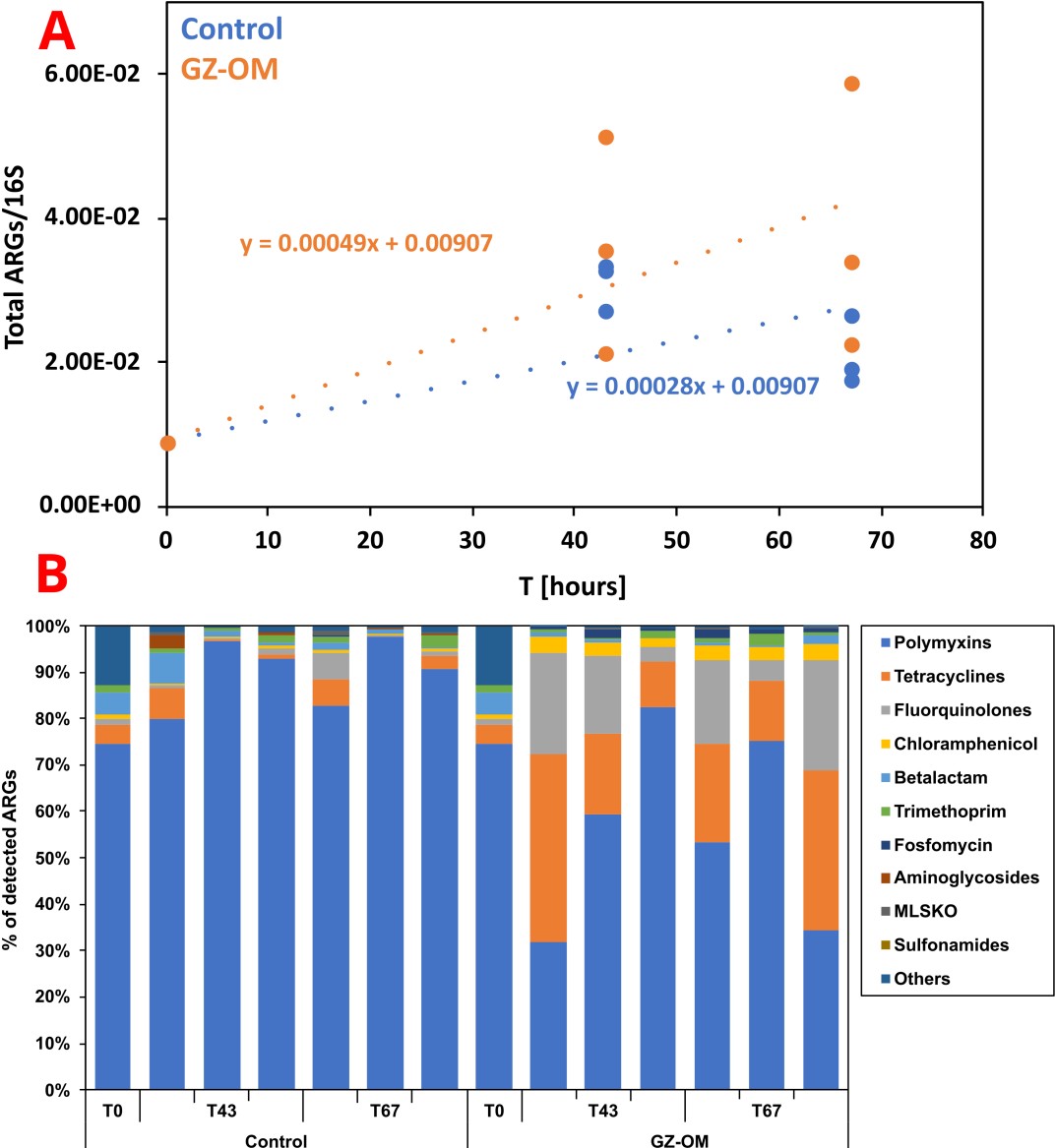

**FIG 2** Resistome dynamics in the *M. leidyi* GZ-OM and control microcosms over time. (A) Pearson correlation of total ARGs per 16S rRNA gene in the metagenomes over time. (B) Proportion of antibiotic resistance gene classes in the total ARGs of the metagenomes (MLSKO = macrolide, lincosamide, streptogramin, ketolide, and oxazolidinone).

community was again significantly different (all $P < 0.05$, AMOVA) (Fig. 4B). Phylogenetic diversity explained a significant proportion of the observed ARG diversity using Procrustes analysis (Correlation index = 0.8019, sum of squares = 0.366, $P = 0.0001$). Consequently, we aimed to identify those bacteria responsible for this shift in the resistome.

## Bacterial genera associated with GZ-OM degradation including human and animal pathogens are potential ARG hosts

To identify which bacteria are responsible for the observed increase in ARGs in the GZ-OM treatment, we analyzed the microbial community compositions. On the phylum level, most notably, in both treatments, the abundance of the most prevalent *Pseudomonadota* increased compared to the initial community from 69.38% to 79.92 ± 3.97% (control) and 81.67 ± 2.99% (GZ-OM) (Fig. 5A) which was significant for the GZ-OM

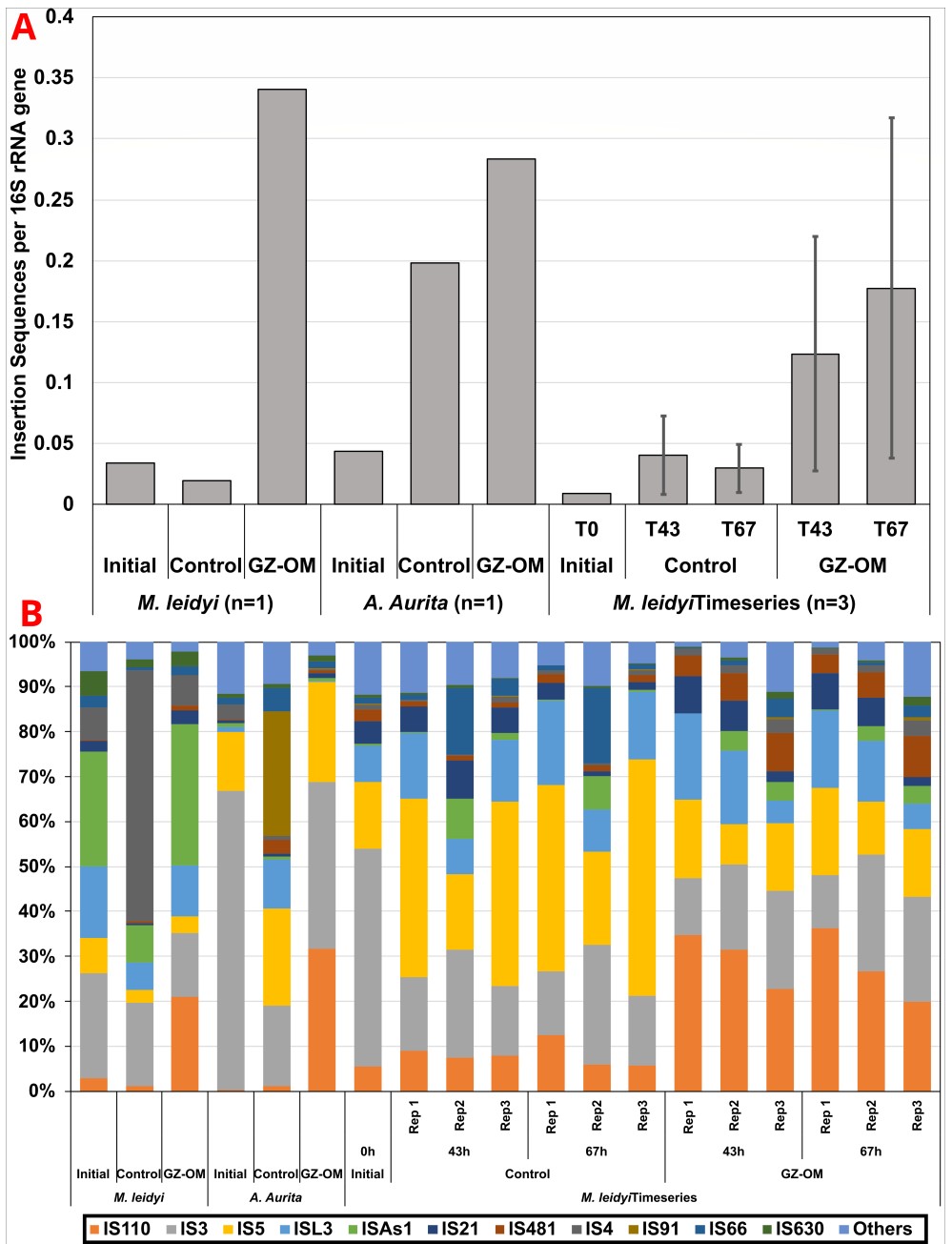

**FIG 3** Mobile genetic element dynamics in the GZ-OM and control microcosm metagenomes. (A) Relative abundance of insertion sequences per 16S rRNA gene. In the *M. leidyi* time series experiment (*n* = 3), averages with error bars depicting standard deviation are shown. (B) Proportion of IS families among the total IS content in the metagenomes.

treatment [$P$ = 0.0498, Friedman's $\chi^2(2)$ =6], while not statistically significant but strongly indicated for the control [$P$ = 0.097, Friedman's $\chi^2(2)$ =4.67]. No significant differences between the GZ-OM and the control treatment were observed and for other phyla, no clear significant trends were detected. However, since genus level diversity analysis suggested that differences between the control and the GZ-OM treatment exist (Fig. 4A), we identified those genera that were highly abundant in general (>0.5% rel. abundance) and more than fivefold significantly elevated in the GZ-OM treatment compared to the initial community as well as the control treatment (all $P$ < 0.05, ANOVA). These probably contributed to the observed increase in ARG levels. The majority of the identified GZ-OM

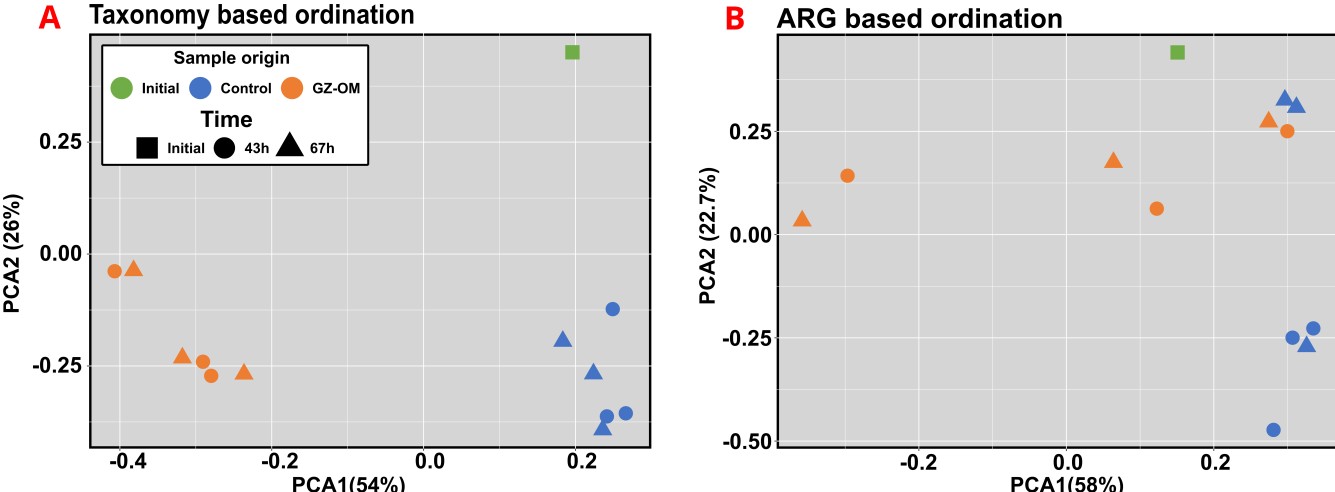

**FIG 4** Microbial community and ARG dynamics in the microcosm experiments. (A) PCA plot of genus-level microbial community composition based on Euclidean distances after Hellinger transformation of the data. (B) PCA plot of individual ARG-level resistome diversity of the microbiomes based on Euclidean distances.

treatment-associated genera belonged to the phylum *Pseudomonadota*. Among these, the most prevalent were *Vibrio* (4.07 ± 3.09%), *Pseudoalteromonas* (3.13 ± 0.60%), and *Thalassotalea* (2.11 ± 0.09%), all previously identified as capable GZ-OM degraders (7, 13). In addition, two typical marine genera were identified: *Colwellia* (0.69 ± 0.08%), associated with a psychrophilic lifestyle (69), and *Algicola* (0.53% ± 0.25%), a regular colonizer of algal surfaces (70). Still, similar to the preliminary data set analyzed, genera regularly hosting human pathogens (67) such as *Enterobacter* (1.37 ± 0.35%) or the aforementioned *Vibrio,* and animal pathogens (71, 72) such as *Arsenophonus* (1.13 ± 0.29) and *Pajaroellobacter* (0.63 ± 0.26%) were identified as GZ-OM associated.

To identify whether those GZ-OM degradation-associated genera are indeed linked with ARGs as their potential hosts we performed a network analysis between ARG and genera abundance in the entire data set revealing a high number of positive genera-genera correlations. Still, when extracting exclusively those 31 connections of ARGs with genera (Fig. 5B) it became apparent, that exclusively the nine previously identified GZ-OM degradation-associated genera could be identified as ARG hosts based on network analysis. Each of these genera had at least one significant connection to an ARG and displayed on average 3.44 ± 2.30 connections to ARGs. Out of the 9 ARGs for which host connections could be inferred, seven displayed connections to multiple hosts, indicating their potential mobility, which is of concern considering that both marine and human/animal pathogen-associated genera were identified to be GZ-OM-associated. Among the identified host genera, *Vibrio* displayed the highest number of connections with seven individual ARGs. Moreover, *Vibrio* was the lone potential host of quinolone ARG *qnr*S6, while sharing host association of quinolone ARG *qnr*S2 and tetracycline ARG *tet* (33) with *Pajaroellobacter*. That is particularly relevant, as we demonstrated that tetracycline and quinolone ARGs displayed the highest increase in GZ-OM treatments.

## Analysis of *Vibrio* isolates from the GZ-OM degradation experiment

As previously stated, *Vibrio* spp. have a highly relevant role not only as degraders but also as potential sources of ARGs and their horizontal dissemination. We consequently used a *Vibrio* isolate (A06) from a previous GZ-OM degradation microcosm experiment to analyze whether this highly enriched GZ-OM degradation-associated genus contains ARGs and MGEs that would display the potential of increased horizontal gene transfer of ARGs during the degradation of GZ-OM. Hence, the isolated *Vibrio* strain A06 was sequenced and annotated. The 16Sr RNA gene of A06 aligned with an average identity of

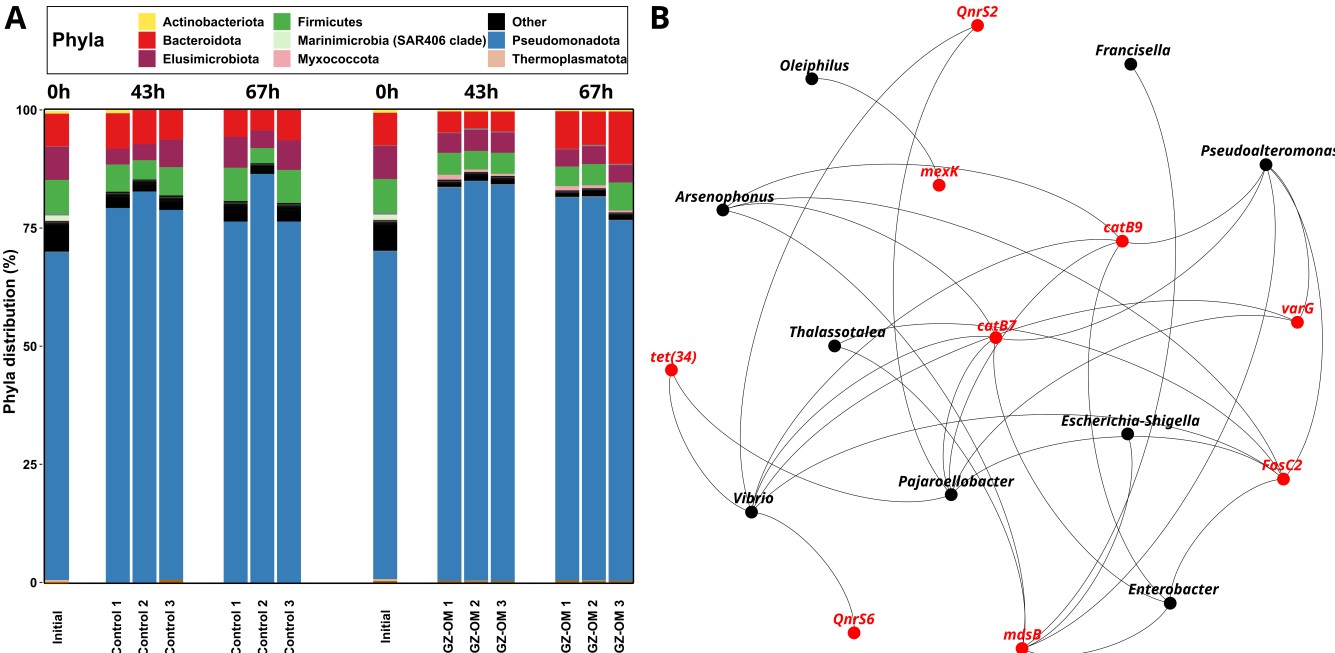

**FIG 5** Microbial community composition and potential ARG hosts in the GZ-OM degradation microcosm experiments. (A) Community composition at the phylum level over time. Phyla with abundance below 1% are grouped as others. (B) Network analysis displaying the correlations in relative abundance between bacterial genera and identified ARGs across all time points and control as well as GZ-OM treatment samples based on significant ($P < 0.05$) Spearman correlations with positive correlation coefficients of $\rho > 0.75$.

99.9% to *Vibrio splendidus* sequences. The HSP60 gene alignment verified that isolate A06 clustered with the species *Vibrio splendidus* with an average identity of 98.3%.

The assembled *V. splendidus* A06 genome consisted of three circular structures of 3.7, 2.1, and 0.17 Mbps with normalized depths/copy numbers of 1×, 0.98×, and 1.58×. While the first two structures matched several *Vibrio* species' first and second chromosomes, the third contained plasmid-related structures. A total of 31 AMR-related genes were distributed across the first two genetic structures (16 in Chromosome 1 and 15 in Chromosome 2), most of which (29/31) coded for multidrug efflux pumps including *mdt*L/A/K/C/H, *mep*A, *emr*D/E, *cat1*, *bmr*A, and *nor*M. The two remaining predicted ARGs were the tetracycline resistance gene *tet34* (Chromosome 1) and a putative *qnr* gene (Chromosome 2), typically associated with diminished susceptibility towards fluoroquinolones. The latter was closely related to *qnr*S sequences from RefSeq (Bootstrap > 0.99). Moreover, 38 transposases belonging mostly to IS families *3*, *4*, and *110* were found in Chromosome 1. Chromosome 2 harbored 23 additional transposases, the majority of which belonged to IS*4* and IS*110* families. Insertion sequences from these two families have been observed in more than 200 different bacterial and archaeal species (73) and have been associated with ARG mobility in environmental bacterial communities (74) and human clinical isolates (75, 76). In the case of A06, no evidence of ARGs or virulence factors within a range of ISs belonging to these families was found. However, several composite transposons harbored hypothetical proteins, suggesting a potentially active role in the incorporation of exogenous genetic material.

The third assembled circular genetic structure was compatible with a plasmid: *rep*A, a *tra* region (incomplete), a partition system (*par*AB), and a toxin/antitoxin system (*phd/doc*) were identified. A 662 bp origin of replication (*ori*C) was annotated immediately downstream from the predicted *rep*A gene. This replicase gene could not be assigned to any incompatibility group present in the PlasmidFinder database, instead, it showed 100% coverage and 98.23% identity with the *rep*A of another non-typed plasmid recovered from *Vibrio crassostreae* (AN: AP025478.1). The structure of the plasmid harbored by A06 showed little similarity with the sequences present in the

NCBI database with two *Vibrio* genetic structures (AP025478.1 and KP795560.1) being the closest relatives.

Several ABC-type efflux pumps were detected on the plasmid, most of these related to inorganic toxin efflux. Moreover, the plasmid carried an ABC pump involved in bacteriocin elimination, flanked by two insertion sequences (IS*150* and IS*Sod8*). While not directly related to AMR, the gene *hipA* was detected on the plasmid. The product of this gene HipA has been reported to cause reversible "dormancy" of *E. coli*, slowing down cell growth and favoring a persister phenotype that is notoriously tolerant to drugs like β-lactams (77, 78). Similar to the chromosomes, a considerable amount of 22 insertion sequences was detected, the most common families being IS*6* and IS*5*. The former is a well-studied family that has major implications in the clinic and is generally related to the incorporation of promoter sequences that could upregulate flanking genes (79). In summary, based on an individual *Vibrio* isolate, we identified the genetic potential for the mobilization and transfer of ARGs through plasmid and IS structures in GZ-OM degrading microbial consortia.

## DISCUSSION

With this study, we established the first link between two emerging marine issues, jellyfish blooms and AMR spread, both likely increasing in projected future ocean scenarios (5, 80). Metagenomic analysis of marine microbial communities exposed to GZ-OM confirmed our hypothesis that decaying GZ-blooms represent a yet over-looked hot spot of AMR proliferation in marine environments. Already after 2–4 days of exposure to GZ-OM, we recorded an up to fourfold increase in relative ARG abundance per 16S rRNA gene copy in the degrader communities compared to ambient marine microbiomes. This increase becomes particularly relevant when considering that bacterial production rates due to the nutrient influx through degradable GZ-OM in the otherwise nutrient-poor marine environment can be up to one order of magnitude elevated (5) with absolute bacterial biomass increasing 10- to 100-fold in the microcosms (7, 12, 13), with equal numbers being reported for natural ecosystems exposed to GZ-OM after bloom events (81). Combining this increase in absolute bacterial and relative ARG abundances, GZ-blooms are predicted to result in an absolute increase of ARG abundance by several orders of magnitude compared to the surrounding marine microbiomes.

The observed trait was consistent, independent of the gelatinous zooplankton species and the year of the experiment, suggesting that the underlying mechanism of this increase in AMR is based on the general influx of nutrients and colonizable surfaces through GZ-OM. Still future work should aim at disentangling the individual contributions of these two general mechanisms, which are further supported by the phylogenetic diversity of the colonizing bacterial communities being highly similar across GZ-OM from different species both in our analysis as well as in studies performed in similar regions of the Mediterranean Sea (82–85). It furthermore proved to be a significant explanatory variable for the observed ARG diversity and increased ARG abundance. Potential carriers of these increasing ARGs were consistent in all our data sets and included *Pseudoalteromonas*, *Vibrio*, and *Alteromonas*, known key GZ-OM degraders (7, 8, 13, 84, 85), as well as *Thalassotalea*, *Colwellia* associated with psychrophilic lifestyle (69), *Algicola*, regular colonizers of algal surfaces (70), and other potential degraders of hydrocarbons in marine environment (*Oleiphilus*, *Anaerosinus*). More importantly, when considering the risks associated with the observed enrichment of ARGs, several genera that contain known potential human or animal pathogens (*Enterobacter*, *Escherichia-Shigella*, *Acinetobacter*, *Vibrio*, *Pajaroellobacter*, *Francisella*, and *Arsenophonus*) (67) were identified to not only be significantly increased in relative abundance in the degrading communities but also correlated with specific ARGs as their potential carriers. This is consistent with previous reports that jellyfish-colonizing microbiomes regularly include elevated proportions of potential human pathogenic strains (84, 85).

The simultaneous increased abundances of potential pathogens and ARGs in the degrading communities do not, on their own, immediately translate into an elevated risk if ARGs are not transferred. Our data provides a strong indication that such horizontal acquisition of ARGs by these potential pathogenic strains is indeed taking place.

First, similar to ARGs, the relative abundance of MGEs in the GZ-OM degrading communities was significantly elevated. These ARG-encoding MGEs have the potential to be transferred even to phylogenetically distant bacterial groups (28–31) and horizontal gene transfer rates are particularly elevated when bacterial abundances and activity are high and bacteria have high encounter rates (32, 33, 86, 87) such as in biofilms formed on GZ-OM particles. The observed high connectivity between marine environmental and potentially pathogenic species in the ARG-genera co-occurrence network as co-hosts of specific identical ARGs suggests that this scenario indeed occurs. The feasibility of increased ARG transfer is moreover supported by *Vibrio* and *Alteromonas,* identified as main players during GZ-OM degradation, being well known for their ability to engage in marine horizontal gene transfer through diverse pathways including conjugation (88, 89), transduction (90), or transformation (91, 92). Many *Vibrio* strains are naturally competent and can take up and integrate free DNA through transformation (91). Their ability to obtain such DNA for incorporation through competitive mechanisms (e.g., type VI secretion systems) from other bacteria (91, 93) might play a major role in the high bacterial density scenarios that are found in biofilms formed on GZ-OM particles.

Second, genomic analysis of the *A. aurita*-associated *V. splendidus* strain revealed a high number of insertion sequences, and numerous multidrug-efflux pumps but also the ARGs *tet* (33) and *qnrS* carried in the two identified chromosomes. The quinolone resistance encoding *qnr* genes are generally plasmid-associated (94), while *tet* (33) has been observed in a broad range of environmental hosts (95), suggesting its general mobility. In addition, a yet unknown plasmidic structure was identified that hosts several insertion sequences, a copy of the *hip*A gene able to induce a dormant state that favors the persistence of ARGs (77, 78), and a bacteriocin efflux pump as part of a compound transposon. Together, this provides a strong indication that the GZ-OM degrading communities have members that indeed possess the necessary genomic plasticity that provides a high potential of acting as donors and recipients of horizontally transferable ARGs. This could constitute a significant risk, as these GZ-OM colonizing communities enriched in AMR and potential pathogens that could acquire novel ARGs can hitchhike on these particle surfaces by drifting with ocean currents over long distances in the ocean interior and coastal environments where exposure to marine organisms of higher complexity such as fish, crustaceans or mollusks (e.g., also commercially important groups) and/or humans is likely (35, 36).

When considering gelatinous zooplankton detritus as a hotspot for the marine spread of AMR, it is also likely that living gelatinous zooplankton colonized by bacteria could equally play a role. Here it is additionally relevant to consider their life stage-specific features. During the polyp stage, meroplanktonic gelatinous zooplankton species are mostly found in coastal areas that frequently are highly anthropogenically impacted (e.g., pillars of industrial ports), where they can accumulate different types of pollutants (96–98). These could provide a (co-)selective potential for ARGs of their microbial colonizers (99, 100) while also being in more direct contact with potential colonizers enriched in ARGs (e.g., from wastewater discharged into the ocean (24)).

After polyp strobilation, syphozoan medusae transition into the planktonic stage of their life cycle, with ephyrae developing into the adult medusa stage. These planktonic stages can drift and/or swim with ocean currents over long distances, and in this way represent an overlooked route of AMR (and ARG) to otherwise not impacted environments. Similarly, ctenophores, which spend their entire life cycle in the planktonic stage, can contribute to this dispersal. For instance, invasive species like *Mnemiopsis leidyi*, which invaded many coastal marine ecosystems globally, might present a special threat. During their planktonic life, they are efficient grazers of a significant part of the ocean's planktonic production (2, 101) and can further accumulate nanoparticles,

microplastic debris (102–105), heavy metals, and pollutants (106) which have the potential to (co-)select for ARGs and increase horizontal gene transfer rates of the colonizing microbes (29, 30, 32, 99). These unexplored aspects regarding the spread of AMR in connection with GZ need to be studied and taken into account, especially when considering harvesting GZ for food, fertilizers, medicine, and cosmetics, or considering their use in wastewater treatment applications (107, 108).

Still, it needs to be taken into consideration that the here employed short-term microcosm experiments may not fully replicate natural scenarios, but they can serve as an important first step toward understanding the complex interactions that occur in marine ecosystems exposed to GZ-blooms. These controlled experiments provide valuable insights and form a foundation for future studies under more natural conditions. Moreover, such future studies could employ deeper sequencing and long-read-based techniques as well as novel, molecular, PCR-based techniques (109, 110) to gain further insights into ARG-host and ARG-MGE associations during GZ-OM degradation.

In conclusion, we here provide evidence that jellyfish blooms are a quintessential "One Health" issue where decreasing environmental health is immediately connected to benign effects on human health by amplifying the spread of antimicrobial resistance genes and their potential transfer to human pathogens. This is of particular relevance as both issues are likely to increase in importance with current climate change projections.

## ACKNOWLEDGMENTS

We acknowledge Eduard Fadeev for conducting preliminary analysis on some of the metagenomic datasets used in the study.

U.K. and T.U.B. were supported by the Explore-AMR and the JPIAMR SEARCHER project funded by the Bundesministerium für Bildung und Forschung under grant numbers 01DO2200 & 01KI2404A. A.X.E., U.K., and T.U.B. were supported by the ACRAS-R project funded by Bundesministerium für Bildung und Forschung under grant number 16GW0355. P.F. was supported through the China Scholarship Council (CSC) under grant number 202004910327. This project received funding from the European Union's Horizon 2020 Research and Innovation Program under the Marie Skłodowska-Curie Grant Agreement No. 793778. G.J.H. was funded by the Austrian Science Fund (FWF) project I04978. T.T. was supported by the Slovenian Research Agency under grant number ARRS J7-2599 and by the Slovenian Research Agency (Research Core Funding No. P1-0237). Exchange between the German and Slovenian groups was supported through the JELLY-AMR Project funded by the DAAD (Project ID: 57747282) and the Public Agency for Scientific Research and Innovation of the Republic of Slovenia (ARIS) (Project ID: B|-DE/ 25-27-001). Responsibility for the information and views expressed in the manuscript lies entirely with the authors.

## AUTHOR AFFILIATIONS

[1]Institute of Hydrobiology, Technische Universität Dresden, Dresden, Germany
[2]Marine Biology Station Piran, National Institute of Biology, Piran, Slovenia
[3]Tsinghua Shenzhen International Graduate School, Institute of Environment and Ecology, Tsinghua University, Shenzhen, China
[4]Department of Functional and Evolutionary Ecology, Bio-Oceanography and Marine Biology Unit, University of Vienna, Vienna, Austria
[5]NIOZ, Department of Marine Microbiology and Biogeochemistry, Royal Netherlands Institute for Sea Research, Den Burg, the Netherlands
[6]Vienna Metabolomics & Proteomics Center, University of Vienna, Vienna, Austria

## AUTHOR ORCIDs

Neža Orel  http://orcid.org/0000-0001-5939-7258
Tinkara Tinta  http://orcid.org/0000-0001-6740-8973

Uli Klümper ⓘ http://orcid.org/0000-0002-4169-6548

## FUNDING

| Funder | Grant(s) | Author(s) |
|---|---|---|
| Bundesministerium für Bildung und Forschung (BMBF) | 01DO2200, 01KI2404A | Peiju Fang |
| | | Thomas U. Berendonk |
| | | Uli Klümper |
| Bundesministerium für Bildung und Forschung (BMBF) | 16GW0355 | Alan X. Elena |
| | | Thomas U. Berendonk |
| | | Uli Klümper |
| China Scholarship Council (CSC) | 202004910327 | Peiju Fang |
| H2020 Marie Skłodowska-Curie Actions | 793778 | Tinkara Tinta |
| Austrian Science Fund (FWF) | I04978 | Gerhard J. Herndl |
| Javna Agencija za Raziskovalno Dejavnost RS (ARRS) | ARRS J7-2599, P1-0237 | Neža Orel |
| | | Tinkara Tinta |
| DAAD | Project ID: 57747282 | Uli Klümper |
| Public Agency for Scientific Research and Innovation of the Republic of Slovenia (ARIS) | Project ID: B\|-DE/25-27-001 | Tinkara Tinta |

## AUTHOR CONTRIBUTIONS

Alan X. Elena, Conceptualization, Data curation, Formal analysis, Investigation, Methodology, Validation, Visualization, Writing – original draft, Writing – review and editing | Neža Orel, Data curation, Formal analysis, Investigation, Validation, Writing – review and editing | Peiju Fang, Formal analysis, Funding acquisition, Investigation, Methodology, Visualization, Writing – review and editing | Gerhard J. Herndl, Funding acquisition, Project administration, Resources, Supervision, Writing – review and editing | Thomas U. Berendonk, Funding acquisition, Investigation, Resources, Supervision, Writing – review and editing | Tinkara Tinta, Conceptualization, Data curation, Formal analysis, Funding acquisition, Investigation, Methodology, Project administration, Resources, Supervision, Validation, Visualization, Writing – original draft, Writing – review and editing | Uli Klümper, Conceptualization, Data curation, Formal analysis, Funding acquisition, Methodology, Project administration, Resources, Supervision, Validation, Visualization, Writing – original draft, Writing – review and editing

## DATA AVAILABILITY

The data sets supporting the conclusions of this article are included within the article or available through the corresponding authors upon reasonable request. Original sequencing data are available in the NCBI Sequence Read Archive under project accession numbers PRJEB63998, PRJEB77875 (*M. leidyi* experiments), PRJNA633735 (*A. aurita* experiments), and PRJEB77200 (*Vibrio splendidus* A06 whole-genome sequencing data).

## ADDITIONAL FILES

The following material is available online.

Open Peer Review

**PEER REVIEW HISTORY (review-history.pdf).** An accounting of the reviewer comments and feedback.

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
