## [Reviewer comments · mSystems]

Jellyfish blooms - an overlooked hotspot and potential vector for the transmission of antimicrobial resistance in marine environments

Alan Elena, Neza Orel, Peiju Fang, Gerhard Herndl, Thomas Ulrich Berendonk, Tinkara Tinta, and Uli Klümper

Corresponding Author(s): Uli Klümper, Technische Universität Dresden

Review Timeline:

Submission Date:	July 30, 2024
Editorial Decision:	October 25, 2024
Revision Received:	December 3, 2024
Accepted:	January 17, 2025

Editor: Jacqueline Goordial

Reviewer(s): Disclosure of reviewer identity is with reference to reviewer comments included in decision letter(s). The following individuals involved in review of your submission have agreed to reveal their identity: Tamar Guy-Haim (Reviewer #2)

Transaction Report:

DOI: <https://doi.org/10.1128/msystems.01012-24>

Re: mSystems01012-24 (Jellyfish blooms - an overlooked hotspot and potential vector for the transmission of antimicrobial resistance in marine environments)

Dear Dr. Uli Klümper:

Both reviewers agreed that the manuscript was of interest for the journal, and acceptable for publication if minor modifications were made. Below you will find their specific comments for revision.

Revision Guidelines

Sincerely,
Jacqueline Goordial
Editor
mSystems

Reviewer #1 (Comments for the Author):

This study tests a novel and interesting hypothesis about the impacts of gelatinous zooplankton (GZ) population collapse on human health: that organic matter resulting from declining GZ blooms may increase the prevalence antibiotic resistance genes (ARGs) and mobile genetic element involved in interbacterial gene transfer of ARGs and potentially other genes related to

human health (e.g., virulence genes). I was a bit confused by the experimental design at first as some of the data were collected from previously conducted and published experiments (Tinta et al 2023 and Fadeev et al 2024, and maybe Tinta 2012 [see comments]) and reanalyzed here to address a new question, while additional experiments were also conducted to increase statistical power and gain additional insights. Given that "mixed" design, I would suggest the Methods section (which explains this well) follow the Introduction and precede the Results, and perhaps leading with a short summary of this overall design. I also think this study, at least for the new experiments conducted which represent a nice time series, could benefit from additional microbiome analyses like ANCOMBC or LEFSE- these are the correct approaches to determine enrichment of microbial taxa across groups and I think the results will be interesting. The paper is well-written (especially the results), the authors provide a very good background describing the relevance and importance of this study, and the conclusions are thought-provoking. I have included additional details for each section below.

Introduction:

Line 54: First paragraph is very long and could be split into 2...A good place to split it could be at "Due to a combination of life history traits..."

Line 76: this paragraph is also very long and could be split

Line 78: I suggest saying "many" instead of "all". Also remove comma from after marine ecosystems in that sentence

Line 303: I question whether comparing your *Vibrio* isolate to other *vibrio* 16S sequences is the best way to determine that it is *splendidus* since 16S sequences are a notoriously poor way to identify *vibrios* (see the extensive literature on using HSP60 to resolve *vibrio* species rather than 16S). I think HSP60 sequences would be better, or alternatively you could align your genome to another published *splendidus* genome and report that similarity as supporting your conclusion drawn from the 16S sequences. If it is low similarity, that would also be worth a deeper look. See also my note on figure 6 below.

Line 378: Also emphasize (and cite) that these taxa are known to engage in HGT...there are many examples of this for *Alteromonas* and *vibrio*, and this is in fact why these organisms are often used as laboratory models to study marine microbiology. I think adding a bit about this would strengthen your findings- since you don't show what organisms the ARGs and MGEs actually belong to I think it would be good to at least demonstrate that from previous research it is very believable that these dominant members would engage in the HGT you are examining

Discussion Other: You might suggest some future studies, for example conducting deeper sequencing so you can determine the microbial hosts of the ARGs you identify as enriched. Additionally, I would discuss the fact the conjugative plasmids are not the only method for accumulating ARGs and that other mechanisms might be contributing to your observations. For example, *vibrios* can be naturally competent and obtain DNA through competitive mechanisms (eg T6SS) could be taking up ARGs from the GZ-OM "bonanza" (example: <https://www.sciencedirect.com/science/article/pii/S1369527421001284>)

Line 507: The reference for this (13) is Tinta et al 2023, not the 2012 paper cited as the original source of the *Vibrio* isolate. I am not sure if this was a typo, but if the isolate was from 2012 (12 years ago), I think a brief justification should be included for why you analyzed the genome of an "old *vibrio*" rather than one from one of the recent 2 studies. Either way please clarify.

Figures:

Figure 6: I don't think Figure 6 adds much to the paper, and it possibly distracts or brings up additional concerns unrelated to your main point (see 16S discussion above) which is to use your isolate to illustrate and bolster your other findings about ARGs and MGEs. If it is included it should be supplemental

Figure 7: I would suggest removing this from the paper or moving this to supplemental as it doesn't really support and perhaps distracts from your main points.

Reviewer #2 (Comments for the Author):

The manuscript by Elena et al. discusses the jellyfish blooms as a potential vector for antimicrobial resistance using short-term experimental microcosms.

The study included a metagenomic re-analysis of two former published experiments on OM generated from the scyphozoan *Aurelia aurita* and the ctenophore *Mnemiopsis leidyi*, which was pooled per treatment ($n=1$) and hence lacked statistical power. To overcome this, the authors performed an additional microcosm experiment using *M. leidyi* with triplicates. While this is acceptable, it must be made clear already in the Abstract that this study combines a re-analysis and a new experiment.

The topic of this study is novel and timely, and the work seems justified and robust. Yet, I had some concerns regarding the statistical analyses of the time evolution experiment. First, such experiments of repeated measures are not independent, and

thus a repeated-measure test should be employed rather than a t-test that was used to compare between the control and the GZ-OM in each time point, separately (eg, repeated measure ANOVA).

Second, the sample size is too small for testing using a parametric t-test (or ANOVA), which assume normality and homogeneity of variances - which were not reported in the ms (impossible in case of small n). Instead you can use a non-parametric repeated measurement test, eg Friedman's Test. A similar problem (which cannot be easily resolved) is with Wilcoxon signed-rank test, which was used to compare between the different classes although $n=1$. The problem here again is the non-independence of the classes, and potentially dependencies that violate the assumptions of this test.

The Introduction lacks a general overview of AMR and ARGs, probably assuming that the readership of mSystems is already familiar with these topics. Yet, I would recommend adding a paragraph generally introducing this topic, providing more examples from the marine environment, showing how these are linked to ocean health.

The experiment was performed in a closed microcosm system with a preserved GZ-OM material and lasted 67 hours. While this setting allows for investigating of short-term bacterial dynamics, it also a caveat. Would you expect to find the same dynamics in an open natural system?

Finally, this paper would benefit from some closer proof reading. It includes numerous grammatical errors, missing genus/species italicization and misplaced words.

Minor comments:

[1] L21 - there is only little evidence that GZ detritus is introduced to the ocean's interior, but rather sinks on the seafloor.

Suggest replacing or adding "... (GZ-OM) sink to the seafloor"

[2] L31 - remove "indicating that" or "suggesting that"

[3] L33,34 - *Vibrio* should appear in italics

[4] L37 - AMR appears here for the first time, provide the full term

[5] L41 - Jellyfish blooms are far from seem exclusively problematic for bathing, as there have been numerous studies on their impact on fisheries, marine food webs, coastal installations. Suggest rephrasing this statement to "in the context of human health, ..."

[6] L44 - change "investigates based on re-analysis" to "used re-analyses to investigate..."

[7] L55 - remove "There", add "total" before "biovolume"

[8] L55 - add citation for GZ being 30% of the total biovolume

[9] L58 - use phylum names consistently. If "ctenophores, salps" then "chaetognaths" and not "Chaetognatha"

[10] L68 - replace "distorts" with "transforms"

[11] L107 - GZ-OM abbreviation is already mentioned

[12] L124 - Wilcoxon signed-rank test was used to compare between the different classes although $n=1$. The problem here again is the non-independence of the classes, and potentially dependencies that violate the assumptions of this test.

[13] L163 - Fig. 1 caption add info on the number of exposure days.

[14] L180 - there seems to be a significant increase in ARGs/16S in the control between $t=0$ and $t=43$ and a decrease in $t=67$. As the analysis used was correlation and linear regression, this was not discussed. Can you suggest an explanation?

Also, do you assume that the ARGs will stay in a steady state if the experiment would have lasted longer?

[15] L185-197: as mentioned, t-tests are not appropriate here

[16] L217-219: move to Discussion

[17] L220-223: where is this data on ISVisp2/6 coming from? If not derived directly from the experiment, move to Discussion

[18] L225: error bars denote std?

[19] L235: replace "Figure 3A" with "Figure 4A"

[20] L237-241: Fig 4 missing A and B indications

[21] L280-294: Can you run the network analysis on the control as well to compare with the GZ-OM?

[22] L296: change "...spp. has" to "have" and "a degrader" to "degraders"

[23] L374: this statement is not applicable in all systems, especially in coastal waters

[24] L390: "..., do not, on their own,..." or rephrase

[25] L413: unclear "higher marine organisms" - did you mean higher diversity of marine organisms? Or higher abundance?

[26] L415-431: nice hypothesis! It would definitely be interesting to try and assess the role of different stages of jellyfish. Yet, there is a bit of confusion in this paragraph. After strobilation, the scyphozoan polyp produces ephyrae which develop into medusae - both stages can drift in the water column. Ctenophores are not medusae not have such a stage, therefore should not appear in the same sentence with polyps/ephyrae/medusae.

mSystems01012-24 General comments

The manuscript by Elena et al. discusses the jellyfish blooms as a potential vector for antimicrobial resistance using short-term experimental microcosms.

The study included a metagenomic re-analyses of two former published experiments on OM generated from the scyphozoan *Aurelia aurita* and the ctenophore *Mnemiopsis leidyi*, which was pooled per treatment (n=1) and hence lacked statistical power. To overcome this, the authors performed an additional microcosm experiment using *M. leidyi* with triplicates. While this is acceptable, it must be made clear already in the Abstract that this study combines a re-analysis and a new experiment.

The topic of this study is novel and timely, and the work seems justified and robust. Yet, I had some concerns regarding the statistical analyses of the time evolution experiment. First, such experiments of repeated measures are not independent, and thus a repeated-measure test should be employed rather than a t-test that was used to compare between the control and the GZ-OM in each time point, separately (eg, repeated measure ANOVA).

Second, the sample size is too small for testing using a parametric t-test (or ANOVA), which assume normality and homogeneity of variances – which were not reported in the ms (impossible in case of small n). Instead you can use a non-parametric repeated measurement test, eg Friedman's Test. A similar problem (which cannot be easily resolved) is with Wilcoxon signed-rank test, which was used to compare between the different classes although n=1. The problem here again is the non-independence of the classes, and potentially dependencies that violate the assumptions of this test.

The Introduction lacks a general overview of AMR and ARGs, probably assuming that the readership of mSystems is already familiar with these topics. Yet, I would recommend adding a paragraph generally introducing this topic, providing more examples from the marine environment, showing how these are linked to ocean health.

The experiment was performed in a closed microcosm system with a preserved GZ-OM material and lasted 67 hours. While this setting allows for investigating of short-term bacterial dynamics, it also a caveat. Would you expect to find the same dynamics in an open natural system?

Finally, this paper would benefit from some closer proof reading. It includes numerous grammatical errors, missing genus/species italicization and misplaced words.

Minor comments:

[1] L21 – there is only little evidence that GZ detritus is introduced to the ocean's interior, but rather sinks on the seafloor. Suggest replacing or adding "... (GZ-OM) sink to the seafloor"

[2] L31 – remove "indicating that" or "suggesting that"

[3] L33,34 – *Vibrio* should appear in italics

[4] L37 – AMR appears here for the first time, provide the full term

[5] L41 – Jellyfish blooms are far from seem exclusively problematic for bathing, as there have been numerous studies on their impact on fisheries, marine food webs, coastal installations. Suggest rephrasing this statement to "in the context of human health, ..."

[6] L44 – change "investigates based on re-analysis" to "used re-analyses to investigate..."

[7] L55 – remove "There", add "total" before "biovolume"

[8] L55 – add citation for GZ being 30% of the total biovolume

[9] L58 – use phylum names consistently. If "ctenophores, salps" then "chaetognaths" and not "Chaetognatha"

[10] L68 – replace "distorts" with "transforms"

[11] L107 – GZ-OM abbreviation is already mentioned

[12] L124 – Wilcoxon signed-rank test was used to compare between the different classes although $n=1$. The problem here again is the non-independence of the classes, and potentially dependencies that violate the assumptions of this test.

[13] L163 – Fig. 1 caption add info on the number of exposure days.

[14] L180 – there seems to be a significant increase in ARGs/16S in the control between $t=0$ and $t=43$ and a decrease in $t=67$. As the analysis used was correlation and linear regression, this was not discussed. Can you suggest an explanation?

Also, do you assume that the ARGs will stay in a steady state if the experiment would have lasted longer?

[15] L185-197: as mentioned, t-tests are not appropriate here

[16] L217-219: move to Discussion

[17] L220-223: where is this data on *ISVisp2/6* coming from? If not derived directly from the experiment, move to Discussion

[18] L225: error bars denote std?

[19] L235: replace “Figure 3A” with “Figure 4A”

[20] L237-241: Fig 4 missing A and B indications

[21] L280-294: Can you run the network analysis on the control as well to compare with the GZ-OM?

[22] L296: change “..spp. has” to “have” and “a degrader” to “degraders”

[23] L374: this statement is not applicable in all systems, especially in coastal waters

[24] L390: “.., do not, on their own,..” or rephrase

[25] L413: unclear “higher marine organisms” – did you mean higher diversity of marine organisms? Or higher abundance?

[26] L415-431: nice hypothesis! It would definitely be interesting to try and assess the role of different stages of jellyfish. Yet, there is a bit of confusion in this paragraph. After strobilation, the scyphozoan polyp produces ephyrae which develop into medusae – both stages can drift in the water column. Ctenophores are not medusae not have such a stage, therefore should not appear in the same sentence with polyps/ephyrae/medusae.

Reviewer #1 (Comments for the Author):

This study tests a novel and interesting hypothesis about the impacts of gelatinous zooplankton (GZ) population collapse on human health: that organic matter resulting from declining GZ blooms may increase the prevalence antibiotic resistance genes (ARGs) and mobile genetic element involved in interbacterial gene transfer of ARGs and potentially other genes related to human health (e.g., virulence genes).

Response: We thank the reviewer for their positive assessment and the valuable comments that have helped us to further clarify and improve the manuscript.

I was a bit confused by the experimental design at first as some of the data were collected from previously conducted and published experiments (Tinta et al 2023 and Fadeev et al 2024, and maybe Tinta 2012 [see comments]) and reanalyzed here to address a new question, while additional experiments were also conducted to increase statistical power and gain additional insights. Given that "mixed" design, I would suggest the Methods section (which explains this well) follow the Introduction and precede the Results, and perhaps leading with a short summary of this overall design.

Response: We thank the reviewer for this valuable suggestion and fully agree that moving the methods section ahead of the results improves the manuscript. We have accordingly moved the methods ahead and now (according to a comment by reviewer #2) additionally mention the experimental design in the abstract to ensure that it is sufficiently clear

I also think this study, at least for the new experiments conducted which represent a nice time series, could benefit from additional microbiome analyses like ANCOMBC or LEFSE- these are the correct approaches to determine enrichment of microbial taxa across groups and I think the results will be interesting.

Response: We had indeed performed additional LEFSE analysis on the dataset prior to the submission of the manuscript. However, due to the limited number of samples combined with the high number of detected genera, the LEFSE analysis only resulted in a very limited number of genera (2: *Deferribacterota* & *Hadarchaeota*) significantly associated with only the GZ-OM treatments. Upon closer inspection, these genera were not detected in the initial microbiome or the control and were very lowly abundant ($<10^{-4}$) in some but not all of the GZ-OM treatments. This means that while these were significantly associated with the GZ-OM treatments, they are unlikely to play a major role and could even be statistical artifacts based on detection limits. Consequently, we decided not to include the LEFSE analysis which usually performs far better on larger datasets in the final version of the manuscript as no relevant insights could be gained from it.

The paper is well-written (especially the results), the authors provide a very good background describing the relevance and importance of this study, and the conclusions are thought-provoking. I have included additional details for each section below.

Response: We thank the reviewer for their positive comment.

Introduction:

Line 54: First paragraph is very long and could be split into 2...A good place to split it could be at "Due to a combination of life history traits..."

Response: We have split the paragraph according to the reviewer's suggestion.

Line 76: this paragraph is also very long and could be split

Response: We have also split this paragraph according to the reviewer's request.

Line 78: I suggest saying "many" instead of "all". Also remove comma from after marine ecosystems in that sentence

Response: We agree and have modified it accordingly.

Line 303: I question whether comparing your *Vibrio* isolate to other *vibrio* 16S sequences is the best way to determine that it is *splendidus* since 16S sequences are a notoriously poor way to identify *vibrios* (see the extensive literature on using HSP60 to resolve *vibrio* species rather than 16S). I think HSP60 sequences would be better, or alternatively you could align your genome to another published *splendidus* genome and report that similarity as supporting your conclusion drawn from the 16S sequences. If it is low similarity, that would also be worth a deeper look. See also my note on figure 6 below.

Response: We have now in addition to the 16S analysis performed a similar analysis on HSP60 sequences and got to the identical conclusion:

Material and Methods:

*"The taxonomic assignment of isolate *Vibrio* A06 was first carried out using the classical 16S rRNA DNA approach. For this, 16S rRNA gene sequences were extracted using Barrnap V0.9(55), concatenated and aligned to 69 available complete *Vibrio* genomes (Taxid: 662) retrieved from the NCBI database and Enterobase(56) using clustalW(57). Previous studies indicate that *Vibrio* species can be more accurately determined by comparison of the heat shock protein 60 (HSP60)(58, 59). Therefore, as an additional identification strategy, we compared the HSP60 sequence of A06 using a similar approach as for the 16S rRNA gene."*

Results:

*"The 16Sr RNA gene of A06 aligned with an average identity of 99.9% to *Vibrio splendidus* sequences. The HSP60 gene alignment verified that isolate A06 clustered with the species *Vibrio splendidus* with an average identity of 98.3%."*

Line 378: Also emphasize (and cite) that these taxa are known to engage in HGT...there are many examples of this for *alteromonas* and *vibrio*, and this is in fact why these organisms are often used as

laboratory models to study marine microbiology. I think adding a bit about this would strengthen your findings- since you don't show what organisms the ARGs and MGEs actually belong to I think it would be good to at least demonstrate that from previous research it is very believable that these dominant members would engage in the HGT you are examining.

Additionally, I would discuss the fact the conjugative plasmids are not the only method for accumulating ARGs and that other mechanisms might be contributing to your observations. For example, vibrios can be naturally competent and obtain DNA through competitive mechanisms (eg T6SS) could be taking up ARGs from the GZ-OM "bonanaza" (example: <https://www.sciencedirect.com/science/article/pii/S1369527421001284>)

Response: This is a great suggestion. We have incorporated a paragraph regarding this into our discussion:

"The feasibility of increased ARG transfer is moreover supported by Vibrio and Alteromonas, identified as main players during GZ-OM degradation, being well-known for their ability to engage in marine horizontal gene transfer through diverse pathways including conjugation(89, 90), transduction(91) or transformation(92, 93). Many Vibrio strains are naturally competent and can take up and integrate free DNA through transformation(92). Their ability to obtain such DNA for incorporation through competitive mechanisms (e.g., type VI secretion systems) from other bacteria (92, 94) might play a major role in the high bacterial density scenarios that are found in biofilms formed on GZ-OM particles."

Discussion Other: You might suggest some future studies, for example conducting deeper sequencing so you can determine the microbial hosts of the ARGs you identify as enriched.

Response: We have added a short sentence regarding this in the end of the discussion:

"Moreover, future studies could employ deeper sequencing and long-read-based techniques as well as novel, molecular, PCR-based techniques (99, 100) to gain further insights into ARG-host and ARG-MGE associations during GZ-OM degradation."

Line 507: The reference for this (13) is Tinta et al 2023, not the 2012 paper cited as the original source of the Vibrio isolate. I am not sure if this was a typo, but if the isolate was from 2012 (12 years ago), I think a brief justification should be included for why you analyzed the genome of an "old vibrio" rather than one from one of the recent 2 studies. Either way please clarify.

Response: The Vibrio genome we analyzed was indeed isolated during the degradation experiment described in Tinta et al. (2012). We did not isolate any bacteria during the experiments described in Tinta et al. (2023), Fadeev et al. (2024) or this manuscript. However, once we realized that *Vibrio* were a key species involved and that close relatives within the same taxonomic group are present in our metagenomic and metaproteomic datasets from Tinta et al. (2023) and Fadeev et al. (2024), we decided to analyze the previously gained isolate from a similar GZ-OM degradation experiment in detail. We believe that this analysis further supports the relevance of the selected isolate as a representative of key gelatinous zooplankton (GZ) degraders and the potential role *Vibrio* plays in the spread of AMR in GZ-OM degrading communities. We also now give this explanation in the manuscripts material and methods section:

“Vibrio were identified as a key bacterial group involved in GZ-OM degradation and the spread of AMR in this study. No isolation of bacteria was performed during the experiments described in (7, 13) or in this manuscript. However, we had access to a Vibrio A06 isolate from a previous GZ-OM degradation experiment with Aurelia aurita biomass described in detail in Tinta et al., 2012(51) that upon whole genome analysis was identified to be closely related to organisms within the same taxonomic groups present in our metagenomic datasets. Consequently, the Vibrio A06 isolate’s whole genome sequence was analyzed in this study to gain further genomic insights into the role Vibrio might play in the spread of AMR during GZ-OM degradation: ”

Figures:

Figure 6: I don't think Figure 6 adds much to the paper, and it possibly distracts or brings up additional concerns unrelated to your main point (see 16S discussion above) which is to use your isolate to illustrate and bolster your other findings about ARGs and MGEs. If it is included it should be supplemental

Response: We have removed Figure 6 from the manuscript according to the reviewer’s suggestion and slightly shortened the respective results part.

Figure 7: I would suggest removing this from the paper or moving this to supplemental as it doesn't really support and perhaps distracts from your main points.

Response: We have removed Figure 7 from the manuscript according to the reviewer's suggestion and slightly shortened the respective results part.

Reviewer #2 (Comments for the Author):

The manuscript by Elena et al. discusses the jellyfish blooms as a potential vector for antimicrobial resistance using short-term experimental microcosms.

Response: We thank the reviewer for their positive comments and valuable insights that have helped us to clearly improve the statistical scrutiny and overall manuscript of our study.

The study included a metagenomic re-analysis of two former published experiments on OM generated from the scyphozoan *Aurelia aurita* and the ctenophore *Mnemiopsis leidyi*, which was pooled per treatment (n=1) and hence lacked statistical power. To overcome this, the authors performed an additional microcosm experiment using *M. leidyi* with triplicates. While this is acceptable, it must be made clear already in the Abstract that this study combines a re-analysis and a new experiment.

Response: We now clearly mention this in the abstract of our study:

*“To test this, we first re-analyzed metagenomes from two previous studies that experimentally evolved marine microbial communities in the presence and absence of OM from *Aurelia aurita* and *Mnemiopsis leidyi* recovered from bloom events and thereafter performed additional time-resolved GZ-OM degradation experiments to improve sample size and statistical power of our analysis. “*

The topic of this study is novel and timely, and the work seems justified and robust. Yet, I had some concerns regarding the statistical analyses of the time evolution experiment. First, such experiments of repeated measures are not independent, and thus a repeated-measure test should be employed rather than a t-test that was used to compare between the control and the GZ-OM in each time point, separately (eg, repeated measure ANOVA). Second, the sample size is too small for testing using a parametric t-test (or ANOVA), which assume normality and homogeneity of variances - which were not reported in the ms (impossible in case of small n). Instead you can use a non-parametric repeated measurement test, eg Friedman's Test. A similar problem (which cannot be easily resolved) is with Wilcoxon signed-rank test, which was used to compare between the different classes although n=1. The problem here again is the non-independence of the classes, and potentially dependencies that violate the assumptions of this test.

Response: We have now replaced the t-tests with Friedman's test according to the reviewer's suggestion since we upon further inspection of the data fully agree with their assessment of t-tests not being appropriate for analysis of the present dataset with repeated measurements. Results of Friedman's test have now been added to the manuscript. While this led to changes in the presented numbers, significant differences previously stated were largely also observed after applying Friedman's test. Only for two tests p values from Friedman's test ranged around 0.1 while being previously significant meaning that a potential effect is indicated but not statistically significant. These two cases were a) the control against the initial community regarding total Pseudomonadota abundance and b) relative tetracycline resistance gene abundance in the GZ-OM treatment vs the initial community. However, none of these changes in significance affect the overall storyline and conclusions presented in the manuscript.

For the Wilcoxon signed-rank test, we agree with the reviewer's assessment that non-independence is possible and have hence removed a statistical evaluation of the data and now rather qualitatively describe it as e.g. *"For the majority of ARG classes relative abundances were consistently increased in the GZ-OM treatment compared to the initial microbiome (10 out of 11 ARG classes) and the control treatment (9 out of 11 ARG classes, Figure 1B)."*

The Introduction lacks a general overview of AMR and ARGs, probably assuming that the readership of mSystems is already familiar with these topics. Yet, I would recommend adding a paragraph generally introducing this topic, providing more examples from the marine environment, showing how these are linked to ocean health.

Response: We have added a corresponding short paragraph to the introduction to introduce the readership to AMR and its marine dimensions:

"Antimicrobial resistance (AMR) and the spread of ARGs is one of the major global health challenges(14) with globally already 4.71 million deaths associated and 1.14 million deaths directly attributable to bacterial AMR in 2021(15). To mitigate the predicted future rise in these numbers it is important to understand AMR evolution, selection and transmission within and across all interconnected "One Health" compartments (humans, animals and the environment)(16, 17). Especially, understanding the biotic and abiotic drivers underlying this spread is crucial to creating targeted intervention measures(18, 19). With abundance of ARGs increasing in many ecosystems due to anthropogenic activities(20, 21), marine ecosystems and their microbiomes are no exception(22, 23). In particular, coastal zones as a likely entry point of ARG-carrying microbes from, for example, wastewater effluents to the marine environments are in the spotlight as they provide exposure points to humans that are using them recreationally(24). For example, increased colonization of marine surfers with AMR bacteria has previously been proven(25). Moreover, various pollutants of marine ecosystems ranging from chemicals to microplastics can contribute to the spread of AMR in non-coastal marine ecosystems which can accumulate in marine animal microbiomes and subsequently through the food chain be conveyed back to terrestrial animals and humans(26). Yet, potential links between these two emerging issues of anthropogenically impacted marine zones, bloom-forming gelatinous zooplankton species, and ARG have not been explored."

The experiment was performed in a closed microcosm system with a preserved GZ-OM material and lasted 67 hours. While this setting allows for investigating of short-term bacterial dynamics, it also a caveat. Would you expect to find the same dynamics in an open natural system?

Response: We agree that studying the response of bacterial communities to fresh inputs of gelatinous zooplankton (GZ)-derived detritus under in situ conditions would be an ideal scenario. However, this approach remains methodologically challenging due to several factors:

- The elusive and patchy nature of GZ populations in natural systems.
- Difficulties in ensuring that sampling and experiments exclusively capture the response of natural microbial communities to a single source of organic matter, i.e., GZ-derived detritus
- The complexity of disentangling interactions between microbes and GZ-derived detrital matter in natural samples, which include a diverse array of biological entities and environmental variables.

While we acknowledge that short-term microcosm experiments may not fully replicate natural scenarios, they serve as an important first step toward understanding the complex interactions that occur in marine ecosystems. These controlled experiments provide valuable insights and form a foundation for future studies under more natural conditions, which we are currently working towards.

In our unpublished data, we conducted a mesocosm experiment to examine the effects of GZ-derived detritus on both phytoplankton and bacterioplankton communities over several days. We found that the bacterial community response to GZ enrichment was highly reproducible. At the same time, we believe that extending incubation times in small, enclosed systems would be even less representative of natural conditions and could lead to well-documented "bottle effect" issues.

At this point we would hence prefer to not speculate about the outcomes, but have added a sentence acknowledging the fact that our results are 'only' from short term experiments in the discussion:

"Still, it needs to be taken into consideration that the here employed short-term microcosm experiments may not fully replicate natural scenarios, but they can serve as an important first step towards understanding the complex interactions that occur in marine ecosystems exposed to GZ-blooms. These controlled experiments provide valuable insights and form a foundation for future studies under more natural conditions."

Finally, this paper would benefit from some closer proof reading. It includes numerous grammatical errors, missing genus/species italicization and misplaced words.

Response: We carefully went through the entire manuscript to improve the language to the best of our ability.

Minor comments:

[1] L21 - there is only little evidence that GZ detritus is introduced to the ocean's interior, but rather sinks on the seafloor. Suggest replacing or adding "... (GZ-OM) sink to the seafloor"

Response: Added as suggested.

[2] L31 - remove "indicating that" or "suggesting that"

Response: Done.

[3] L33,34 - *Vibrio* should appear in italics

Response: Here and throughout the manuscript we ensured that gene names and species names are properly italicized.

[4] L37 - AMR appears here for the first time, provide the full term

Response: We replaced AMR with ARG here which was previously defined to limit the amount of abbreviations used in the abstract.

[5] L41 - Jellyfish blooms are far from seem exclusively problematic for bathing, as there have been numerous studies on their impact on fisheries, marine food webs, coastal installations. Suggest rephrasing this statement to "in the context of human health, ..."

Response: We have adjusted the phrasing per the suggestion of the reviewer.

[6] L44 - change "investigates based on re-analysis" to "used re-analyses to investigate..."

Response: We have rephrased it accordingly.

[7] L55 - remove "There", add "total" before "biovolume"

Response: Done.

[8] L55 - add citation for GZ being 30% of the total biovolume

Response: The citation is the same one as used for the second part of the sentence.

[9] L58 - use phylum names consistently. If "ctenophores, salps" then "chaetognaths" and not "Chaetognatha"

Response: This has now been made consistent according to the correction supplied by the reviewer.

[10] L68 - replace "distorts" with "transforms"

Response: Done.

[11] L107 - GZ-OM abbreviation is already mentioned

Response: It has been amended accordingly.

[12] L124 - Wilcoxon signed-rank test was used to compare between the different classes although n=1. The problem here again is the non-independence of the classes, and potentially dependencies that violate the assumptions of this test.

Response: We agree, please refer to our response above regarding the statistical analysis.

[13] L163 - Fig. 1 caption add info on the number of exposure days.

Response: Added: *"after 21 h for M. leidy and 32 h for A. aurita"*

[14] L180 - there seems to be a significant increase in ARGs/16S in the control between t=0 and t=43 and a decrease in t=67. As the analysis used was correlation and linear regression, this was not discussed. Can you suggest an explanation? Also, do you assume that the ARGs will stay in a steady state if the experiment would have lasted longer?

Response: Statistically there was no significant difference between t=43 and t=67 for either the control or the GZ-OM treatment. An explanation could be that at that point the carrying capacity in the microcosm experiment was already reached. However, with the limited amount of samples and time points this is rather speculative. In the future, we will run similar experiments with far higher temporal resolution (see response to the comment above regarding the duration of the experiment) to resolve these issues further.

[15] L185-197: as mentioned, t-tests are not appropriate here

Response: We have changed it. See above.

[16] L217-219: move to Discussion

Response: We have moved this sentence.

[17] L220-223: where is this data on ISVisp2/6 coming from? If not derived directly from the experiment, move to Discussion

Response: We have rephrased the sentence in question to clarify that this is derived from the experiments.

[18] L225: error bars denote std?

Response: Indeed. This has now been added to the figure legend.

[19] L235: replace "Figure 3A" with "Figure 4A"

Response: This has been corrected.

[20] L237-241: Fig 4 missing A and B indications

Response: A and B have been added to the figure.

[21] L280-294: Can you run the network analysis on the control as well to compare with the GZ-OM?

Response: Network analysis already includes all samples from all time points of control and GZ-OM treatments. This is now clarified in the figure legend.

[22] L296: change "..spp. has" to "have" and "a degrader" to "degraders"

Response: Done.

[23] L374: this statement is not applicable in all systems, especially in coastal waters

Response: We have qualified the statement to reflect that this is based only studies of similar regions in the Mediterranean sea.

[24] L390: ".., do not, on their own,..." or rephrase

Response: Changed.

[25] L413: unclear "higher marine organisms" - did you mean higher diversity of marine organisms? Or higher abundance?

Response: This has been changed to *"marine organisms of higher complexity such as fish, crustaceans or mollusks"* to clarify what we understand under higher marine organisms.

[26] L415-431: nice hypothesis! It would definitely be interesting to try and assess the role of different stages of jellyfish. Yet, there is a bit of confusion in this paragraph. After strobilation, the scyphozoan polyp produces ephyrae which develop into medusae - both stages can drift in the water column. Ctenophores are not medusae not have such a stage, therefore should not appear in the same sentence with polyps/ephyrae/medusae.

Response: We agree that the original formulation of the paragraph could have been clearer. We acknowledge that ctenophores are not jellyfish and do not have a polyp stage, so they should not be grouped in the same sentence when discussing hypotheses about early cnidarian jellyfish life stages, such as ephyrae being propagated to different locations via currents.

That said, this distinction does not preclude the possibility that ctenophores, at any life stage, can similarly be transported by currents to new locations. To address this, we have now revised the paragraph to improve clarity and ensure the distinction between these taxa is properly conveyed:

"After polyp strobilation, scyphozoan medusae transition into the planktonic stage of their life cycle, with ephyrae developing into the adult medusa stage. These planktonic stages can drift and/or swim with ocean currents over long distances, and in this way represent an overlooked route of AMR (and ARG) to otherwise not impacted environments. Similarly, ctenophores, which spend their entire life cycle in the planktonic stage, can contribute to this dispersal. For instance, invasive species like Mnemiopsis leidyi, which invaded many coastal marine ecosystems globally, might present a special threat. During their planktonic life, they are efficient grazers of a significant part of the ocean's planktonic production(2, 91) and can further accumulate nanoparticles, microplastic debris(92–95), heavy metals and pollutants(96)

which have the potential to (co-)select for ARGs and increase horizontal gene transfer rates of the colonizing microbes(24, 25, 27, 88). These unexplored aspects regarding the spread of AMR in connection with GZ need to be studied and taken into account, especially when considering harvesting GZ for food, fertilizers, medicine, and cosmetics, or considering their use in wastewater treatment applications(97, 98).“

Re: mSystems01012-24R1 (Jellyfish blooms - an overlooked hotspot and potential vector for the transmission of antimicrobial resistance in marine environments)

Dear Dr. Uli Klümper:

Thank you for the revised manuscript. Both reviewers have found the corrections made to be acceptable and have requested no further modifications.

Your manuscript has been accepted, and I am forwarding it to the ASM production staff for publication. Your paper will first be checked to make sure all elements meet the technical requirements. ASM staff will contact you if anything needs to be revised before copyediting and production can begin. Otherwise, you will be notified when your proofs are ready to be viewed.

Sincerely,
Jacqueline Goordial
Editor
mSystems

Reviewer #1 (Comments for the Author):

I am satisfied with the authors revisions and I think this will be a valuable contribution to the field.

Reviewer #2 d I am overall satisfied with the corrections made by the authors.